# How Effective Is Conservation Biological Control in Regulating Insect Pest Populations in Organic Crop Production Systems?

**DOI:** 10.3390/insects11110744

**Published:** 2020-10-29

**Authors:** Raymond A. Cloyd

**Affiliations:** Department of Entomology, Kansas State University, Manhattan, KS 66506, USA; rcloyd@ksu.edu; Tel.: +1-785-532-4750; Fax: +1-785-532-6232

**Keywords:** conservation biological control, intraguild predation, floral resources, population regulation, parasitoids, predators, weed diversity

## Abstract

**Simple Summary:**

Organic crop production systems typically rely on conservation biological control to increase and sustain natural enemies including parasitoids and predators that will regulate insect pest populations below damaging levels. The use of flowering plants or floral resources to attract and retain natural enemies in organic crop production systems has not been consistent, based on the scientific literature, and most importantly, many studies do not correlate an increase in natural enemies with a reduction in plant damage. This may be associated with the effects of intraguild predation or the negative effects that can occur when multiple natural enemies are present in an ecosystem. Consequently, although incorporating flowering plants into organic crop production systems may increase the natural enemy assemblages, more robust scientific studies are warranted to determine the actual effects of natural enemies in reducing plant damage associated with insect pest populations.

**Abstract:**

Organic crop production systems are designed to enhance or preserve the presence of natural enemies, including parasitoids and predators, by means of conservation biological control, which involves providing environments and habitats that sustain natural enemy assemblages. Conservation biological control can be accomplished by providing flowering plants (floral resources) that will attract and retain natural enemies. Natural enemies, in turn, will regulate existing insect pest populations to levels that minimize plant damage. However, evidence is not consistent, based on the scientific literature, that providing natural enemies with flowering plants will result in an abundance of natural enemies sufficient to regulate insect pest populations below economically damaging levels. The reason that conservation biological control has not been found to sufficiently regulate insect pest populations in organic crop production systems across the scientific literature is associated with complex interactions related to intraguild predation, the emission of plant volatiles, weed diversity, and climate and ecosystem resources across locations where studies have been conducted.

## 1. Introduction

Organic crop production systems have increased worldwide as a consequence of the concern over pesticide (e.g., insecticides, miticides, and fungicides) inputs, which has resulted in a greater demand for organic products [1]. In addition, there is an emphasis on how organic crop production systems can exploit the potential benefits of biological control [2]. However, issues have arisen regarding the effectiveness of conservation biological control in sufficiently regulating insect pest populations below damaging levels in organic crop production systems [3].

This article discusses several topics related to how effective biological control is in regulating insect pest populations in organic crop production systems, including: plant diversity, based on the presence of flowering plants (floral resources), and the regulation of insect pest populations by natural enemies (i.e., parasitoids and predators); intraguild predation; effects of plant volatiles on natural enemies; impact of weed diversity on natural enemies; and the effects of insecticides used in organic crop production systems on natural enemy populations.

## 2. Organic Crop Production Systems

Organic crop production systems use practices, such as conservation biological control, which are designed to promote biological and ecological processes that result in mitigating plant damage without inputs from synthetic insecticides [4]. However, organic crop production systems can experience more problems with multiple insect pest complexes than conventional crop production systems that use insecticides [4]. Therefore, organic crop production systems attempt to conserve natural enemy populations by means of plant diversity through the presence of flowering plants, which are intended to increase the abundance and assemblage of natural enemies including parasitoids and predators [5]. Consequently, organic crop production systems rely almost exclusively on natural enemies to regulate insect pest populations below damaging levels. The reason for relying on natural enemies is primarily because there are only a limited number of insecticides registered for use in organic crop production systems compared to conventional crop production systems [2,6,7] and these contact, short residual insecticides are typically less effective in suppressing insect pest populations than synthetic insecticides [3].

Insect pest management is based on ecological principles and the use of multiple plant protection strategies, including: cultural, physical, insecticidal, and biological [2,3,8,9,10]. Applied biological control is associated with utilizing natural enemies, including parasitoids and predators, which have the potential to regulate insect pest populations at levels that minimize plant damage [11,12]. The fundamental basis of organic crop production systems is to enhance or preserve natural means through conservation biological control and minimize inputs from insecticides.

## 3. Conservation Biological Control

Conservation biological control involves manipulating or preserving an environment or habitat by increasing plant diversity and enhancing the availability of flowering plants that foster the establishment of natural enemy populations, which would otherwise be absent in conventional crop production systems [3,13,14,15,16,17]. Natural enemy populations are intended to thrive and simultaneously regulate insect pest populations below damaging levels without inputs from broad-spectrum synthetic insecticides [18,19].

However, there is an assumption associated with the concept of conservation biological control in that increasing plant diversity, based on the presence of flowering plants in organic crop production systems, is directly correlated with an increased abundance of natural enemies by providing food sources (nectar and pollen), mating sites, and shelter [20,21,22,23]. Consequently, the intended increase in the abundance of natural enemies results in the regulation of insect pest populations [24]. This assumption is based on the premise that natural enemies are influenced by plant diversity, thus leading to an increase in natural enemy diversity (species), which will result in greater parasitism and predation rates [24,25]. In addition, the assumption presumes that increasing the assemblage of natural enemies (number and species) will lead to the enhanced regulation of multiple insect pest complexes and a reduction in plant damage [16].

Adult parasitoids and predators are known to feed on the nectar and pollen of various flowering plants [26]. However, the contribution of flowering plants in attracting and retaining natural enemies in the environment and their direct enhancement in regulating insect pest populations are complex [23,27]. Although evidence supports claims made regarding the benefits of flowering plants and natural enemies in increasing the regulation of insect pest populations [28], there are limited scientific studies that clearly demonstrate that plant diversity, based on the availability of flowering plants, actually results in an increase in natural enemies that provide the sufficient regulation of insect pest populations below economically damaging levels.

## 4. Does Plant Diversity Enhance Regulation of Insect Pest Populations by Natural Enemies?

There are questions associated with plant diversity, based on the presence of flowering plants, and natural enemies, such as:*Does an increase in natural enemy assemblage and abundance lead to enhanced regulation of insect pest populations and a subsequent reduction in plant damage in organic crop production systems? What levels of insect pest numbers are required to sustain natural enemy populations throughout the growing season?*How will natural enemies be affected by different levels of insect abundance? Can natural enemies sufficiently regulate multiple-insect pest complexes and mitigate plant damage?*Can the presence of natural enemies prevent crops from being exposed to viruses transmitted by certain insect pests including aphids, leafhoppers, and whiteflies?

In addition, is there a relationship between plant diversity, affiliated with flowering plants, and the effective regulation of insect pests by natural enemies [24]? To address the questions presented above, we should determine if there is an association between flowering plants presence and natural enemy abundance, the subsequent regulation of insect pest populations, and a reduction in plant damage.

Flowering plants provide a valuable food source (nectar and pollen) for the adult stage of numerous natural enemies (Figure 1) [29,30,31]. Nectar is an important carbohydrate source that is essential for the survival and reproduction of many natural enemies [18], especially parasitoids [32,33]; however, nectar can vary in quantity and quality depending on plant types [17,22,26,34,35,36]. Pollen is also required by many natural enemies, helping to increase female fecundity, and similar to nectar, can vary in quantity and quality among plant types during the growing season [26].

Certain flowering plants, such as sweet alyssum, *Lobularia maritima* (Figure 2), may sustain natural enemies when insect pest populations are low [37]. However, not all flowering plants are attractive to all natural enemies nor do they provide a viable food source [38,39], because natural enemies such as parasitoids cannot exploit the nutrients available from certain flowers due to flower morphology [38,40]. Moreover, insect pests may experience less mortality from natural enemies when feeding on plants in flower than feeding on plants in the vegetation stage due to altering the blend of volatiles that are emitted and attractive to natural enemies [30].

Flowering plants can enhance the survival of natural enemies by serving as a food source (nectar and pollen), as well as provide shelter, mating sites, and refuge, which can increase their abundance and potential to regulate insect pest populations [15,20,23,27,28,31,41,42,43,44]. Flowering plants must be attractive and frequently visited by natural enemies to retain them within the vicinity, and consequently they can regulate existing insect pest populations [17]. In addition, flowering plants may provide a source of alternative prey (hosts) that serve to sustain the adult stage of many different types of natural enemies in the absence of the main insect pests [31,45,46]. However, the presence of alternative prey (hosts) can disrupt the regulation of insect pest populations by natural enemies [47] by distracting natural enemies (e.g., predators) away from the main insect pests [48].

Flowering plants should only provide a benefit to natural enemies and should not be susceptible to different insect pests, which can intensify problems with insect pests on the main crops [23,32,40,49,50,51,52,53,54,55]. Furthermore, flowering plants must bloom early and be available throughout the growing season so a food source will be constantly available to an assemblage of natural enemies that will regulate insect pest populations below damaging levels [56,57]. However, flowering plants can vary widely in their attractiveness and nectar accessibility to natural enemies [26,57]. For example, flower architecture can influence the value of certain flowers, such as sweet alyssum, *L. maritima*, and buckwheat, *Fagopyrum esculentum*, to natural enemies based on the ability of natural enemies to obtain nectar from flowers [7,36,40,49,58,59]. In addition, some flowering plants can repel natural enemies including parasitoids [26]. Table 1 provides a listing of flowering plants that attract certain natural enemies including parasitoids and predators based on scientific studies conducted under laboratory or field conditions.

There are claims that lower insect pest populations in organic crop production systems are a consequence of implementing farming practices that promote natural enemy diversity and abundance [60]. Practices such as cover cropping and intercropping that include the use of flowering plants can result in an increase in the assemblage of natural enemies, thus leading to a reduction in insect pest problems [61]. Studies have shown that plant diversity, based on the abundance of flowering plants, is enhanced in organic crop production systems when cover cropping, which leads to a greater diversity of natural enemy species compared to conventional crop production systems [62]. Consequently, natural enemies will mitigate insect pest outbreaks or regulate insect pest populations [63,64,65].

However, there is conflicting information on whether or not insect pest regulation by natural enemies increases when cover cropping or intercropping practices are used in organic crop production systems [7,16,24,66,67,68]. For example, using a mixture of cover crops, including: purple vetch, *Vicia benghalensis*; barley, *Hordeum vulgare*; fava bean, *Vicia fava*; Austrian winter pea, *Pisum sativum*; and common vetch, *Vicia sativa*, did not result in an increase in egg parasitism by the parasitoid, *Anagrus* spp. (Hymenoptera: Encyrtidae) [69]. Buchanan and Hooks (2018) [70] found that a mixed species of cover crops that included barley, *H. vulgare*; crimson clover, *Trifolium incarnatum*, and a barley, *H. vulgare* + crimson clover, *T. incarnatum* mixture did not result in attracting predators or lead to the regulation of a variety of insect pests. In addition, intercropping coriander, *Coridandrum sativum*, and chrysanthemum, *Chrysanthemum coronarium*, did not lead to a reduction in infestations of the aphid, *Nasonovia ribisnigri* Mosley (Hemiptera: Aphididae) [61].

Extensive robust field studies are lacking, and the results obtained from many studies, both laboratory and semi-field, are either inconclusive or not consistent [15,33,71]. In addition, approaches to managing insect pest populations in organic crop production systems can differ widely among producers based on the location and crops grown [24]. Nonetheless, the presence of flowering plants may not necessarily translate into an increase in parasitism by parasitoids or predation by predators [26]. For example, there was no difference in parasitism by the parasitoid, *Copidosoma aretas* (Walker) (Hymenoptera: Encyrtidae), on populations of the strawberry tortrix, *Acleris comariana* Lienig and Zeller (Lepidoptera: Tortricidae), in the presence of flowering plants [72]. The use of seven different flowering plants resulted in the parasitism of lettuce leafminers (agromyzids); however, the parasitism provided by the six different parasitoids did not translate into the effective regulation of leafminer populations [73]. A study found that the brown lacewing, *Micromus tasmaniae* Walker (Neuroptera: Hemerobiidae), fed upon fewer aphids when flowering buckwheat, *F. esculentum*, was present [74]. The effect of flowering plants on the potential for natural enemies to regulate insect pest populations below damaging levels is complex due to the plant–pest–natural enemy interactions [23,75].

Studies conducted in agricultural and natural settings indicate that an increase in the assemblage of parasitoids and predators can positively or negatively impact attack rates and prey consumption rates, which can influence the ability of natural enemies to sufficiently regulate insect pest populations [24]. Therefore, plant diversity can either impair or promote the abundance of natural enemies and their ability to regulate insect pest populations and more importantly, reduce plant damage [16]. A study reported that in organic lettuce *Lactuca sativa* production, at least four different syrphid species were involved in regulating the populations of the aphid, *N*. *ribisnigra*, below economically damaging levels [76]. Another study found that the parasitism rates of the parasitoid, *Microplitis mediator* Haliday (Hymenoptera: Braconidae), on the cabbage moth, *Mamestra brassicae* Linnaeus (Lepidoptera: Noctuidae), were higher when buckwheat, *F. esculentum*, cornflower, *Centaurea cyanus*, and common vetch, *Vicia sativa*, were present [54]. Tschumi et al. (2016) [77] demonstrated that incorporating 11 different plant species into flower strips near potato, *Solanum tuberosum*, crops resulted in the regulation of aphids by hoverflies, ladybird beetles, and lacewings. However, the study was limited in scope since the study was only conducted from June through August of 2013, and there was no mention of the aphid species collected or the blooming period of the flower strips. Zhao et al. (1992) [75] found that there was an increase in imported cabbageworm, *Pieris rapae* (Linnaeus) (Lepidotpera: Pieridae), and diamondback moth, *Plutella xylostella* (Linnaeus) (Lepidoptera: Plutellidae) larvae, and no difference in parasitism rates by the parasitoid, *Diadegma insulare* (Cresson) (Hymenoptera: Ichneumonidae), regardless of the presence of flowering plants.

There is a general misconception that plant diversity, based on the presence of flowering plants, across all organic crop production systems, leads to a reduction in insect pest populations below damaging levels due to an abundance and diversity of natural enemies [67]. However, the premise is not consistent with empirical data [26,78]. Furthermore, the premise that natural enemies will disperse from flowering plants to the main crop and regulate insect pest populations may not be valid [67,79]. Although the number of natural enemies may increase under organic crop production systems, this may not translate into the sufficient and consistent regulation of insect pest populations [24,80], based on higher predation or parasitism rates [81,82], or a reduction in plant damage.

The effects of plant diversity on natural enemy abundance and subsequent effectiveness in regulating insect pest populations and reducing potential plant damage can vary depending on the cropping system. For example, although non-pest insect diversity was higher in a tomato, *Solanum lycopersicum*, a crop associated with organic compared to conventional cropping systems, pest damage was similar despite insecticide use being lower in the organic crop production system [24]. Drinkwater et al. (1995) [83] reported that fruit and leaf damage caused by a variety of insect pests to tomatoes was not different between organic and conventional crop production systems.

Studies have shown that organic crop production systems may not lead to an increase in parasitism rates despite an increase in the diversity of parasitoids [84]. For example, there was no significant difference in aphid mortality, affiliated with parasitism rate, regardless of parasitoid diversity (based on genera and species) and abundance, between organic (14.7% mortality) and conventional (21.3% mortality) crop production systems [82]. Furthermore, parasitoids and predators may not attack or feed on all insect pests, which would compromise the ability of natural enemies to sufficiently regulate insect pest populations below damaging levels. For instance, the biological control of aphids, including the green peach aphid, *Myzus persicae* (Sulzer) (Hemiptera: Aphididae), and cabbage aphid, *Brevicoryne brassicae* (Linnaeus) (Hemiptera: Aphididae), did not increase as plant diversity increased [16].

There is no evidence indicating that an increase in ground beetle (carabid) or spider (arachnid) abundance or numbers under organic crop production systems leads to sufficient regulation of insect pest populations under field conditions [85]. Although an increase in plant diversity may enhance the abundance of spiders (arachnids) [86], this does not lead to the greater regulation of insect pest populations. Higher numbers of ground beetle (carabid) larvae and adults may be present in the soil or aboveground in organic crop production systems [87]; however, this does not result in the improved regulation of insect pest populations.

Natural enemy species can vary significantly depending on location, which may impact effectiveness in regulating insect pest populations [76]. Moreover, environmental conditions (e.g., temperature, relative humidity, and photoperiod) can vary along with the availability of abundant flowering plants. In addition, flowering plants may not be sufficient to support or promote the establishment of natural enemies [24].

An important ecological interaction in organic crop production systems that can significantly influence the ability of natural enemies to sufficiently regulate insect pest populations below damaging levels, regardless of numbers and species, is intraguild predation.

## 5. Intraguild Predation

Intraguild predation is associated with species that utilize similar, and often limiting, resources resulting in competition [88,89]. However, the interactions that occur with the simultaneous presence of multiple natural enemies (e.g., parasitoids and predators) are complex [90]. Intraguild predation may negatively influence or disrupt the ability of natural enemies to sufficiently regulate insect pest populations [91,92,93,94,95,96,97]. However, intraguild predation does not always result in disrupting the regulation of insect pest populations by natural enemies [5,90,98,99,100].

Competition among natural enemies within an organic crop production system can lead to intraguild predation [101], which can disrupt the capacity of natural enemies to sufficiently regulate insect pest populations by reducing populations of natural enemies [92]. In fact, certain predators may feed on different life stages (eggs, larvae/nymphs, pupae, and adults) of other natural enemies [76]. Hence, lower natural enemy populations can enhance the survival of insect pest populations, because insect pest populations escape exposure to natural enemies. For example, predators such as *Orius* spp. (anthocorids), lacewings (chrysopids), and spiders (arachnids) may prey upon the eggs and/or larvae of hoverflies (syrphids), thus increasing aphid survival because of the reduction in hoverfly populations and reduced feeding on aphids [76].

Green lacewing, *Chrysoperla carnea* Stephens (Neuroptera: Chrysopidae) larvae will prey on the larvae of the convergent lady beetle, *Hippodamia convergens* Guérin-Méneville (Coleoptera: Coccinellidae), and immature stages of the parasitoid, *Aphidius smithi* Sharma and Subba Rao (Hymenoptera: Braconidae), developing inside the pea aphid, *Acyrthosiphon pisum* (Harris) (Hemiptera: Aphididae) [102]. However, green lacewing larvae are themselves preyed upon by a number of predators, including: the leafhopper assassin bug, *Zelus renardii* Kolenati (Hemiptea: Reduviidae); damsel bugs, *Nabis* spp.; and big-eyed bugs, *Geocoris* spp. [91].

Generalist predators such as ground beetles (carabids), rove beetles (staphylinds), ladybird beetles (cocinellids), and spiders (arachnids), can disrupt the ability of parasitoids to regulate aphid populations by feeding on free-living adults, feeding on immature and pupal stages developing inside the prey or host, or feeding on mummified aphids [92,96,99,103,104,105]. For example, nymphs and adults of the spined stilt bug, *Jalysus wickhami* Van Duzee (Hemiptera: Berytidae), feed on the pupal stages of the parasitoid, *Cotesia congregata* (Say) (Hymenoptera: Braconidae) [103]. Press et al. (1974) [106] reported that the predaceous bug, *Xylocoris flavipes* (Reuter) (Hemiptera: Anthocoridae), preyed upon the immature stages of the ectoparasitoid, *Bracon hebetor* Say (Hymenoptera: Braconidae), resulting in an increase in adult densities of the Indianmeal moth, *Plodia interpunctella* (Hübner) (Lepidoptera: Pyralidae).

However, Harvey and Eubanks (2005) [5] showed that the red imported fire ant, *Solenopsis invicta* Buren (Hymenoptera: Formicidae), did not negatively affect the biological control of the diamondback moth, *P. xylostella*, by the parasitoid, *Cotesia plutellae* Kurdjumov (Hymenoptera: Braconidae). Another study found that simultaneously releasing two parasitoids, *Encarsia formosa* Gahan (Hymenoptera: Aphelinidae) and *Encarsia pergandiella* Howard (Hymenoptera: Aphelinidae), and a predator, *Delphastus pusillus* LeConte (Coleoptera: Coccinellidae), did not affect the regulation of the silverleaf whitefly, *Bemisia argentifolii* Bellows and Perring (Hemiptera: Aleyrodidae), populations [90]. These studies indicate that intraguild predation does not always disrupt the ability of natural enemies to regulate insect pest populations.

The presence of multiple insect pests in organic crop production systems can negatively affect biological control, because certain predators will feed on non-target prey instead of the main insect pests [107]. For instance, higher numbers of wolf spiders (lycosids) and ground beetles (carabids) did not improve the regulation of insect pest populations, because the predators preyed upon each other rather than preying upon the main insect pests [108]. The orb-weaving spider, *Metedeira grinnelli* (Coolidge) (Araneidae), was found to displace and prey upon another orb-weaving spider, *Cyclosa turbinate* (Walckenaer) (Araneidae), resulting in reduced predation on insect pests [109]. Regardless, there is evidence that increasing plant diversity can diminish intraguild predation by generalist predators, subsequently leading to the improved regulation of insect pest populations [97].

In addition to intraguild predation, natural enemies may fail to regulate insect pest populations in organic crop production systems due to incompatible and/or asynchronous life histories between insect pest populations and natural enemies, and disruption by resident ant populations [110].

## 6. Effect of Plant Volatiles on Natural Enemies

Plant volatiles are organic compounds associated with the breakdown products of secondary metabolites emitted by leaves and flowers in response to feeding by herbivores and are used by parasitoids and some predators to locate insect pests [30,111,112,113,114,115,116,117,118,119]. However, these plant volatiles, which are complex mixtures [30,120], can vary substantially based on plant species [121,122]. Therefore, plant volatiles can influence the ability of parasitoids to locate and regulate insect pest populations. Floral odors emitted by flowers may directly reduce the attractiveness of certain plant volatiles that are used by parasitoids to locate prey or hosts [123,124], which could impact the ability of parasitoids to regulate insect pest populations in organic crop production systems. In addition, floral odors can reduce parasitism rates, which will allow insect pests to escape regulation by parasitoids [124].

## 7. Impact of Weed Diversity on Natural Enemies

Organic crop production systems may favor or promote a diversity of weed species, which, when in flower, serve as a food source (nectar and pollen) for natural enemies such as hoverflies and other predators [34,125,126,127]. However, weeds can also serve as alternative food sources for certain insect pests such as aphids, whiteflies, and leafhoppers [125,128]. Consequently, the presence of weeds can result in greater insect pest problems [3,125,129], which will negatively affect the ability of natural enemies to sufficiently regulate insect pest populations [130]. Depending on species, flowering weeds may promote greater plant diversity, which can lead to an increase in the assemblage of natural enemies [16,97,129,131]. However, the presence of natural enemies does not translate into an increase in regulating insect pest populations below damaging levels [3,125]. In addition, flowering weeds can vary in attracting natural enemies based on flower morphology and blooming time during the growing season. Furthermore, and more importantly, weeds can serve as reservoirs of diseases (e.g., viruses) and insect vectors including aphids and leafhoppers [128].

## 8. How Do Insecticides Affect Natural Enemies?

Insecticides used in organic crop production systems, compared to those used in conventional crop production systems, are less stable when exposed to ultra-violet light and degrade quickly, which results in shorter residual activity [2,132]. As such, more frequent applications are needed, which can increase the risk of negative effects to natural enemies. However, the timing of insecticide applications can reduce any harmful direct or indirect effects to natural enemies [2]. Nonetheless, some of the insecticides available (botanicals and plant-derived essential oils) can directly harm natural enemies and honey bees [133]. Even materials such as the particle film called kaolin clay, which is used in organic crop production systems, are harmful to natural enemies [134,135,136,137]. Moreover, some insecticides registered for use in organic crop production systems are broad-spectrum, including insecticidal soap (potassium salts of fatty acids) and horticultural oil (mineral-based) which may disrupt the regulation of insect pest populations by natural enemies, thus increasing the potential for secondary pest outbreaks [138].

## 9. Conclusions

The effect of plant diversity, associated with flowering plants (floral resources), in sustaining natural enemies including parasitoids and predators, and in enhancing the regulation of insect pest populations in organic crop production systems, is complex. The scientific literature does not provide consistent evidence that the presence of flowering plants leads to an increase in natural enemies, resulting in the sufficient regulation of insect pest populations and reduced plant damage. The reason is that crop production systems, locations, and the environmental conditions (e.g., temperature, rainfall, and day length) are different across studies.

There are many factors that can influence pest–natural enemy–plant interactions in organic crop production systems, which consequently compromise the conservation biological control and the ability of natural enemy assemblages to regulate insect pest populations below damaging levels, including: types of natural enemies and numbers present, types of flowering plants, blooming time of flowering plants, intraguild predation, and the levels of insect pest abundance.

## Figures and Tables

**Figure 1 insects-11-00744-f001:**
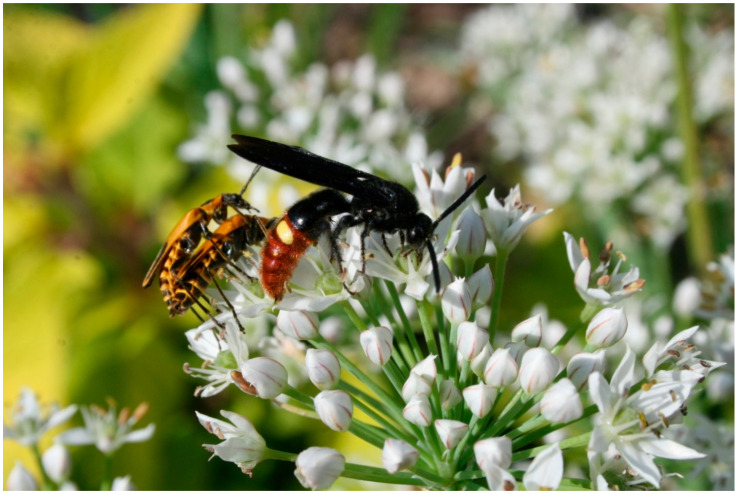
*Scolia dubia* adult feeding on the nectar of wild onion (*Allium* spp.) flower (Raymond Cloyd: Kansas State University; Manhattan, KS, USA).

**Figure 2 insects-11-00744-f002:**
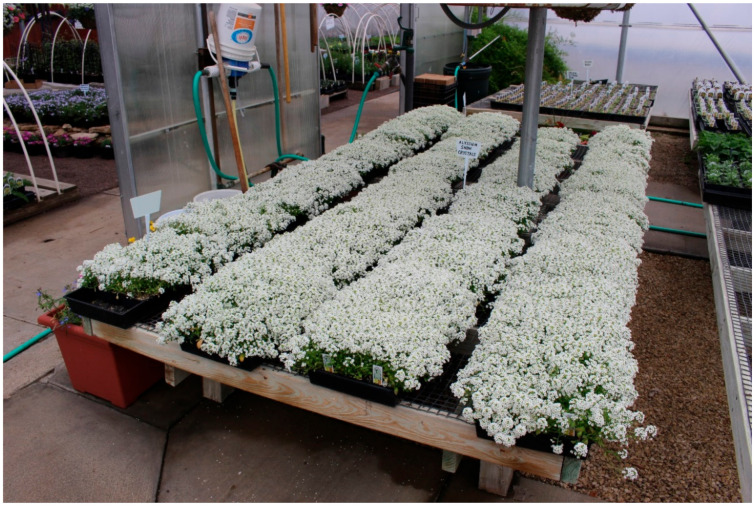
Sweet alyssum, *Lobularia maritima* flowers are attractive to many different types of natural enemies (Raymond Cloyd: Kansas State University; Manhattan, KS, USA).

**Table 1 insects-11-00744-t001:** Flowering plants (common name and scientific name) that attract certain natural enemies.

Flowering Plants	Natural Enemies
Sweet alyssum (*Lobularia maritima*)	Syrphids (hoverflies) [7,42]
	*Orius* spp. [7]
	Coccinellids (ladybird beetles) [7]
	*Trichogramma carverae* Oatman and Pinto (Hymenoptera: Trichogrammatidae) [50]
Buckwheat (*Fagopyrum esculentum*)	Syrphids (hoverflies) [42,50]
	*Trissolcus basalis* (Wollaston) (Hymenoptera: Platygastridae) [17]
	*Microplitis mediator* (Haliday) (Hymenoptera: Braconidae) [54]
Cornflower (*Centaurea cyanis*)	*Microplitis mediator* (Haliday) (Hymenoptera: Braconidae) [54]
Common vetch (*Vicia sativa*)	*Microplitis mediator* (Haliday) (Hymenoptera: Braconidae) [54]
Candytuft (*Iberis amara*)	*Microplitis mediator* (Haliday) (Hymenoptera: Braconidae) [54]
Ground elder (*Aegopodium podagraria*)	*Heterospius prosopidis* (Viereck) (Hymenoptera: Braconidae) [26]
Wild marjoram (*Origanum vulgare*)	*Pimpla turionellae* (Linnaeus) (Hymenoptera: Ichneumonidae) [26]
	*Heterospilus prosopidis* (Viereck) (Hymenoptera: Braconidae) [26]

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
