# Peer review of "How Effective Is Conservation Biological Control in Regulating Insect Pest Populations in Organic Crop Production Systems?"

_insects, 2020, doi:10.3390/insects11110744_

Round 1
Reviewer 1 Report
This manuscript provides a review of conservation biological control in organic crops using floral resources. The information is relevant and useful and should be published. The manuscript is well written, and I have no specific comments or suggestions.
Author Response
First of all, I want to thank the reviewer for taking the time to review the manuscript.
The reviewer had no specific comments or suggestions.
Reviewer 2 Report
The author may explain that when it is refering to insect pest populations it may include some mites pest species.
The author includes serval studies where flowering plant and plant diversity in organic crop production system necessarily do not lead to an increase in levels of parasitism or an increase in parasitoid or predator diversity, but it is not discussed which factors may affect these results?

Author Response
First of all, I want to thank the reviewer for reviewing the manuscript. Below are my responses to the reviewers comments:
- When I first started developing the review I was going to include mites. However, based on my extensive review of the scientific literature I found minimal to no information on how plant diversity influences mite populations.
- I did discuss the factors that may impact why plant diversity does not result in parasitoid or predator diversity, and a subsequent increase in regulating insect pest populations. I mentioned how intraguild predation can influence the diversity of parasitoid and predators. In addition, flowering plant type, floral volatiles, floral resource abundance, and time of year can influence the assemblage of natural enemies. In some cases, no explanations are provided.

Reviewer 3 Report
The submitted review by Cloyd is well motivated, the structure is appropriate, and the manuscript is well written without missing any key details. The conclusions are supported by the results reviewed, indicating that the utility of flowering plants or floral resources to attract and retain natural enemies in organic crop production systems was inconsistent based on the scientific literature. The work further suggests more robust scientific studies are warranted to determine the actual effects of natural enemies in reducing plant damage associated with insect pest populations.
It would be useful for the author to address whether increased natural enemy (e.g., entomopathogen) diversity in soil affects belowground herbivores (i.e., on lines 244-296), given that the choice of specific cover crops may sustain higher levels of entomopathogens in the soil. For example, a more recent study by Milosavljevic et al. 2020 (https://doi.org/10.1016/j.biocontrol.2020.104317) assessed the effects of entomopathogen richness on herbivore suppression and plant productivity in a system consisting of entomopathogens, wireworm herbivores, and wheat plants. The authors found that the composition of entomopathogen communities had stronger effects on a pest species than species richness, suggesting that careful selection of entomopathogen species for biological control may be more impactful than promoting entomopathogen diversity. Adding this information could benefit the discussion.
Overall, a nice piece of work. I enjoyed reading this manuscript.
I hope you will consider revising with what I have noted in mind and resubmit.
Author Response
First of all, I want to thank the reviewer for reviewing the manuscript and providing feedback. The focus of the article was the influence of plant diversity on aboveground natural enemies (e.g. parasitoids and predators) and how they affect or regulate insect pest populations, and reduce plant damage. Discussing the impact of belowground arthropod (insects and mites) diversity on belowground herbivore populations would be a good topic for a separate article.
Furthermore, I did read the article that the reviewer mentioned (The composition of soil-dwelling pathogen communities mediates effects on wireworm herbivores and wheat productivity) and found the following concerns:
a. The study looked at the effects of applying two entomopathogenic fungi and two entomopathogenic nematodes; however, there was no mention of the costs associated with making such applications in the field.
b. The study was conducted in one-year for 6-months. The study should have been conducted longer and for two-years.
c. There was only one harvest date (October 2, 2015). There should have been a series of harvests to assess yield.
d. In addition, there were a number of inconsistencies associated with the study.
I think the article highlights the need to conduct more research on the interactions that occur belowground.